# Preeclampsia, prevalence and associated factors

**Martin Chakulya**[1,2]*, **Prince Mulambo**[1,2], **Gift C. Chama**[1,2], **Lillian Nalavwe**[1,2], **Isaac M. Pulukuta**[1], **Portipher Simwaba**[1,2], **Wana Nyichiwu**[1,2], **Enniless Chilobe**[1,2], **Alice Mwape**[1,2], **Emmanuel Luwaya**[1,2], **Sydney Mulamfu**[1,2], **Chileleko Siakabanze**[1,2], **Katongo H. Mutengo**[1,2], **Lukundo Siame**[1,2], **Joreen P. Povia**[3], **Bridget Namusika**[1], **Bislom C. Mweene**[1], **Annet Kirabo**[4,5,6,7], **Sepiso K. Masenga**[1,2,4,5,7]*

**1** Department of Pathology and Microbiology, Mulungushi University, School of Medicine and Health Sciences, Livingstone, Zambia, **2** Department of Cardiovascular Science and Metabolic Diseases, Livingstone Center for Prevention and Translational Science, Livingstone, Zambia, **3** Department of Health Economics, Livingstone Center for Prevention and Translational Science, Livingstone, Zambia, **4** Department of Medicine, Vanderbilt University Medical Center, Nashville, Tennessee, United States of America, **5** Department of Molecular Physiology & Biophysics, Vanderbilt University, Nashville, Tennessee, United States of America, **6** Vanderbilt Center for Immunobiology, Vanderbilt Institute for Infection, Immunology and Inflammation, Vanderbilt University Medical Center, Nashville, Tennessee, United States of America, **7** Vanderbilt Institute for Global Health, Vanderbilt University Medical Center, Nashville, Tennessee, United States of America

* chakulyamartin1@gmail.com (MC), sepisomasenga@lcpts.org (SKM)

## Abstract

### Background

Preeclampsia (PE) is a significant obstetric complication associated with adverse maternal and fetal outcomes. Zambia, like several Sub-Saharan African nations, experiences a high prevalence of pregnancy-related hypertension, with PE being a major contributor to maternal and foetal mortality. This study aimed to identify the factors associated with the development of PE.

### Methods

We conducted a cross-sectional study at Livingstone University Teaching Hospital in Zambia (LUTH). PE was defined as new-onset hypertension with systolic blood pressure ≥140 mmHg or diastolic blood pressure ≥90 mmHg occurring after 20 weeks of gestation, accompanied by proteinuria with dipstick reading of 1+. Data from patients' most recent hospital visits was collected by trained research assistants using medical record abstraction. A total of 1018 participants were included. Demographic, clinical, and haematological parameters were analysed. We conducted both descriptive and inferential analyses using Stata version 17. Univariable and multivariable logistic regression were employed to investigate factors associated with PE.

**Data availability statement:** The raw data underlying the results presented in the study have been uploaded as supporting information.

**Funding:** The author(s) received no specific funding for this work.

**Competing interests:** The authors have declared that no competing interests exist.

## Results

The median age of participants was 27 years (IQR: 21–33). The prevalence of PE was 12.2% (n = 124). Among the 17.7% (n = 172) of participants who were employed, 19.1% (n = 33) had PE. Factors positively associated with PE included increasing age (Adjusted Odds Ratio [AOR]: 1.07, 95% Confidence Interval [CI]: 1.02–1.12, p = 0.002), a previous history of PE (AOR: 30.8, 95% CI: 8.7–108.6, p < 0.001), and a family history of PE (AOR: 9.97, 95% CI: 2.53–39.27, p = 0.001). Conversely, a unit (week) increase in gestational age was negatively associated with PE (AOR: 0.89, 95% CI: 0.83–0.97, p = 0.007).

## Conclusion

This study identifies maternal age, family history, and prior obstetric history as key associated factors for PE, emphasizing the need for targeted screening and early intervention. Enhanced prenatal care, including routine risk assessments, patient education, and regular monitoring through blood pressure checks, urine protein testing, and fetal growth assessments, is cardinal for early detection and effective management of high-risk individuals.

## Background

Preeclampsia (PE) is a hypertensive disorder of pregnancy and a leading cause of maternal and perinatal morbidity and mortality globally [1]. The burden of PE is disproportionately borne by low- and middle-income countries (LMICs), where health systems are often ill-equipped to manage its severe complications, such as eclampsia, HELLP syndrome, and placental abruption [2–4]. In Sub-Saharan Africa, preeclampsia (PE) is a critical public health crisis, accounting for 56% of the global burden and 9% of maternal deaths in the region [5]. Within Zambia, a previous study at the University Teaching Hospital in Lusaka reported a high prevalence of 12%, aligning with the elevated rates seen across the region and highlighting the significant local burden of this condition [6].

While the global etiology of PE is multifactorial, maternal characteristics, genetics, immunological, environmental, and lifestyle factors collectively contribute to its risk [7–9]. However, much of the existing literature on these associated factors is derived from high-income countries, and the applicability of these models to Zambian populations is not well established. Evidence from similar contexts in Africa highlights the potential importance of factors such as advanced maternal age, high parity, pre-existing hypertension, and family history [10,11]. Furthermore, genetic variants linked to angiogenesis and immune regulation, along with familial clustering, highlight hereditary and epigenetic contributions which can be specific to certain populations [10,12]. However, comprehensive data on these correlates, particularly from specific regions within high-burden countries like Zambia, are scarce.

Understanding the correlates of PE within the Southern Province of Zambia is not only critical for elucidating its aetiology but also for advancing preventive strategies,

early diagnostic tools, and targeted therapeutic interventions for this population. Therefore, this study aimed to determine the prevalence and identify the factors associated with the development of PE at Livingstone University Teaching Hospital to provide much-needed local evidence to inform clinical practice and public health policy.

## Methods

### Study design and site

This study was a cross-sectional study conducted at the obstetrics and gynaecology department at Livingstone University Teaching Hospital (LUTH) among women with and without pre-eclampsia. The department receives referrals for pregnant women from various health facilities within the southern province and western provinces of Zambia.

### Eligibility and recruitment

Data were abstracted from hospital records of pregnant women between 15 and 42 years who attended the Obstetrics and Gynecology (OBGYN) department, following confirmation of pregnancy. Variables of interest, including demographic and clinical information, were extracted from patient records. The records were chosen using a systematic sampling method. The list of patient records was ordered by date of visit. A starting point was then randomly selected before selecting every third file from the list for screening and eligibility. Any record with missing data on age and outcome was excluded from the study. The chosen records were subsequently entered into the Research Electronic Data Capture (REDCap) system.

### Sample size

The estimated minimal sample size was 1018. We estimated a prevalence of 7.7% from a study conducted in Zambia [6], an alpha level of 0.05 (Z = 1.96 for a 95% CI), a margin of error of 5%, and a design effect of 1. Sample size estimation was performed using OpenEpi software. The formula applied was:

$$n = \text{DEFF} \times \frac{N\, Z^2_{1-\frac{\alpha}{2}}\, p(1-p)}{d^2(N-1) +\, Z^2_{1-\frac{\alpha}{2}}\, p(1-p)}$$

where n is the sample size, p is the expected proportion (0.077), d is the desired margin of error (0.05), Z is the Z-score (1.96), and DEFF is the design effect (1).

### Data collection and study procedures

**Research variables and definitions.** PE was defined by clinicians as new-onset hypertension with systolic blood pressure ≥140 mmHg or diastolic blood pressure ≥90 mmHg occurring after 20 weeks of gestation, accompanied by proteinuria with dipstick reading of 1+ [1].HELLP syndrome was defined as a life-threatening pregnancy complication characterized by haemolysis, elevated liver enzymes, and a low platelet count [13]. In this study, PE was the outcome variable. The independent variables were categorized as follows:

Sociodemographic variables: maternal age, employment status, place of residence, marital status, history and status of smoking, and history of alcohol consumption. Obstetric and reproductive history: parity, gravidity, history of preeclampsia (PE), family history of PE, family history of eclampsia, and family history of chronic hypertension.

Clinical variables: proteinuria detected via dipstick, hypertension, severity of PE, HELLP syndrome, placental abruption, complications associated with PE, anaemia, gestational age in weeks, mode of delivery, occurrence of eclampsia, presence of comorbidities, and diabetes.

Laboratory haematological variables: red blood cell count, white blood cell count (leukocytes), monocyte count, lymphocyte count, platelet count, alanine transaminase (ALT), creatinine levels, and haemoglobin levels.

## Data collection

Data from patients' most recent hospital visits were collected by trained research assistants using medical record abstraction between 1st December 2024–30th March 2025. To ensure accuracy and completeness, senior data abstractors subsequently audited the collected information.

## Data analysis

We exported data from the REDCap application to Microsoft Excel 2016 for cleaning and coding. Stata version 17 was then used for data analysis. Descriptive statistics were employed to summarize categorical variables through frequencies and percentages, while continuous variables were summarized using the median and interquartile range. The Shapiro-Wilk test was applied to evaluate data normality. The association between two categorical variables was assessed using the chi-square test, and differences between two medians were evaluated with the Wilcoxon rank-sum test. Univariable and multivariable logistic regression analyses were conducted to identify factors associated with PE. Variables with a p-value < 0.01 in the univariable analysis, along with those of known clinical importance, were considered for inclusion in the multivariable logistic regression model. We excluded variables with significant collinearity. A backward stepwise selection process was used to arrive at the final adjusted model.

## Ethical approval

Ethical approval was obtained from the Mulungushi University School of Medicine and Health Sciences Research Ethics Committee (ethics reference number SMHS-MU2-2024-59) on 05th April 2024 and LUTH administration gave permission to access patient's records. All data collected and analysed were de-identified to ensure complete confidentiality. No identifying information was abstracted or recorded. Secondary data were used in this project thus, written/verbal consent was not applicable and was therefore waived by the ethics committee.

## Results

Out of an initial 3,600 eligible files, 2,582 were excluded due to incomplete data or because they pertained to non-gravid women. This left 1,018 files that met the inclusion criteria. All of these were subsequently analyzed (Fig 1).

## Basic characteristic of the study participants

Table 1 illustrates basic characteristic of the study participants. The median age of the participants was 27 years (IQR: 21, 33). The prevalence of PE was 12.2% (n = 124). Among the 17.7% (n = 172) of employed participants for whom data was available (n = 971), 19.2% (n = 33) had PE. Urban residents had a significantly higher prevalence of PE; among 263 urban women, 38.0% (n = 100) were diagnosed with PE, compared to 2.7% (n = 20) of the 734 rural women. Additionally, among 635 married participants, 14.3% (n = 91) had PE. A history of PE was reported in 6.4% (n = 37) of the 582 participants with available data on this variable, and 75.7% (n = 28) of these women developed PE again in the current pregnancy. Hypertension was observed in 61.5% (n = 123) of the 200 women with a hypertension diagnosis. Among those with a family history of PE (2.6%, n = 26), the majority (80.8%, n = 21) developed PE. Similarly, of the 24 participants (4.3% of 555) with a family history of eclampsia, 75.0% (n = 18) experienced PE. Furthermore, among the 31 women (5.5% of 563) with eclampsia, 51.6% (n = 16) had PE. Conversely, among participants with a family history of chronic hypertension (12.5%, n = 75 of 602), the majority (65.3%, n = 49) did not develop PE. Regarding haematological indices, the median haemoglobin level was 11.5 g/dL (IQR: 10.3, 12.4).

This figure show age and blood count parameters between females with and without PE respectively. **A.** Age: PE group (n = 124) vs. non-PE group (n = 894), **B.** Gestational age: PE group (n = 114) vs. non-PE group (n = 896), **C.** Gestational age at birth: PE group (n = 114) vs. non-PE group (n = 869), **D.** Systolic blood pressure: PE group (n = 124) vs. non-PE

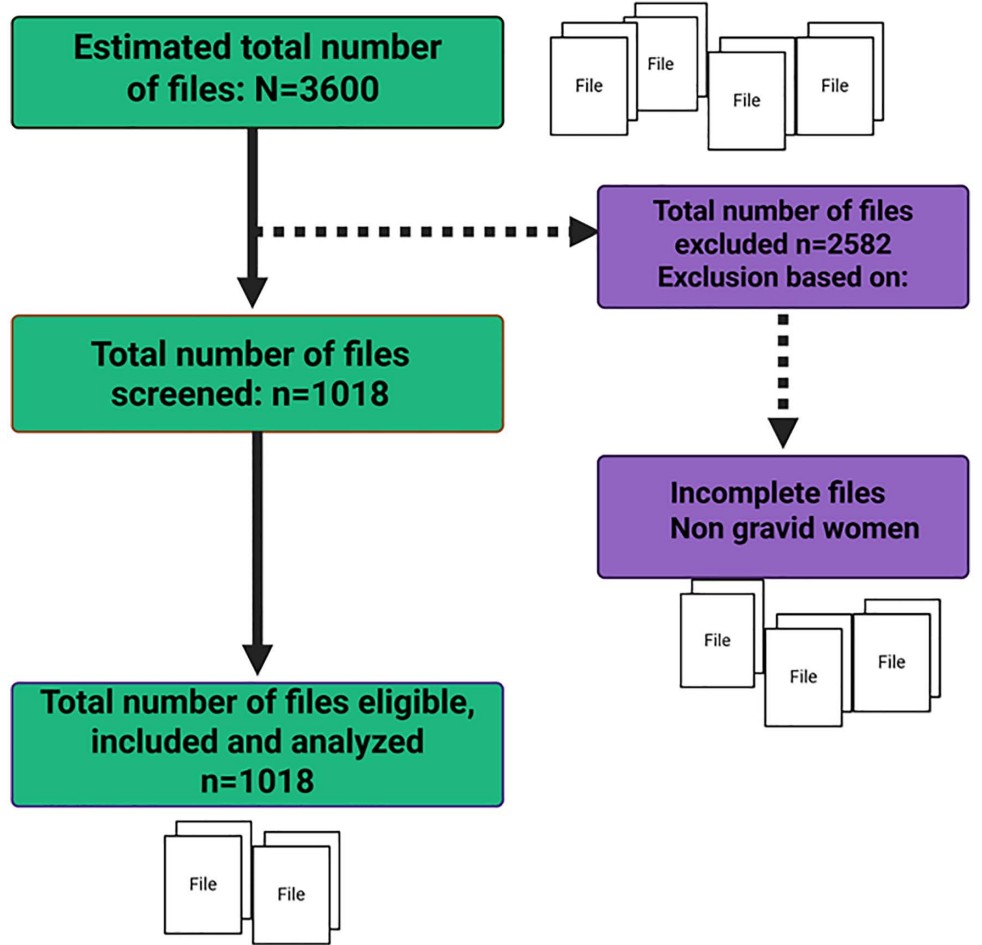

**Fig 1. Participant recruitment flow.**

group (n = 888), **E.** Diastolic blood pressure: PE group (n = 124) vs. non-PE group (n = 888), **F.** Red blood cell count: PE group (n = 69) vs. non-PE group (n = 239), **G.** Leukocyte count: PE group (n = 69) vs. non-PE group (n = 234), **H.** Neutrophil count: PE group (n = 61) vs. non-PE group (n = 211), **I.** Platelet count: PE group (n = 69) vs. non-PE group (n = 234), **J.** Alanine aminotransferase (ALT): PE group (n = 14) vs. non-PE group (n = 31), **K.** Creatinine levels: PE group (n = 11) vs. non-PE group (n = 29), **L.** Lymphocyte count: PE group (n = 63) vs. non-PE group (n = 211), **M.** Haemoglobin (HB): PE group (n = 74) vs. non-PE group (n = 263). $*p < 0.05$; $**p < 0.0.1$; ns, $p > 0.05$; $***p < 0.00$; $****p < 0.001$. See Fig 2.

## Univariable and multivariable analysis of factors associated with PE

Table 2 shows results of univariable and adjusted multivariable analysis of factors associated with PE. In univariable analysis as the age increases, the odds of having PE increased by 1.05 times (OR: 1.05: 95%CI: 1.02, 1.08 $p < 0.001$). An increase in gestational age decreased the odds of having PE by 14% (OR: 0.86: 95%CI: 0.83, 0.89 $p < 0.001$). Additionally for every unit increase in family History of PEthe odds of having PE increased by 36.2. (OR: 36.2: 95% CI: 13.3, 98.1, $P < 0.001$). Furthermore, for every unit increase in a positive history of PE the odds of having PE increased by 28.2. (OR: 28.2: 95%CI: 12.6, 63.0: $p < 0.001$).

**Table 1. Basic characteristics of the study participants.**

| Variables | Total N | Category | Frequency, n (%) | Preeclampsia Yes, n (% of category) | Preeclampsia No, n (% of category) | P- value |
|---|---|---|---|---|---|---|
| **Employment Status** | 971 | Employed | 172 (17.7) | 33 (19.2) | 139 (80.8) | 0.003 |
| | | Unemployed | 799 (82.3) | 87 (10.9) | 712 (89.1) | |
| **Residence** | 997 | Urban | 263 (26.4) | **100 (38.0)** | **163 (62.0)** | 0.01 |
| | | Rural | 734 (73.6) | 20 (2.7) | 714 (97.3) | |
| **Marital Status** | 964 | Married | 635 (65.9) | 91 (14.3) | 544 (85.7) | 0.009 |
| | | Not Married | 329 (34.1) | 28 (8.5) | 301 (91.5) | |
| **History of Preeclampsia** | 582 | Yes | 37 (6.4) | 28 (75.7) | 9 (24.3) | <0.001 |
| | | No | 545 (93.6) | 54 (9.9) | 491 (90.1) | |
| **Proteinuria on Dipstick** | 1018 | Yes | 153 (15.0) | 123 (80.4) | 30 (19.6) | <0.001 |
| | | No | 865 (85.0) | **1 (0.1)** | **864 (99.9)** | |
| **Hypertension** | 1018 | Yes | 200 (19.6) | 123 (61.5) | 77 (38.5) | <0.001 |
| | | No | 818 (80.4) | **1 (0.1)** | **817 (99.9)** | |
| **Preeclampsia Severity** | 124 | Mild | 63 (50.8) | 63 (100.0) | 0 (0.0) | 0.308 |
| | | Severe | 61 (49.2) | 60 (98.4) | 1 (1.6) | |
| **HELLP Syndrome** | 457 | Yes | 3 (0.7) | 2 (66.7) | 1 (33.3) | 0.005 |
| | | No | 454 (99.3) | 56 (12.3) | 398 (87.7) | |
| **Placental Abruption** | 507 | Yes | 7 (1.4) | 1 (14.3) | 6 (85.7) | 0.881 |
| | | No | 500 (98.6) | 62 (12.4) | 438 (87.6) | |
| **Comorbidities** | 427 | Yes | 24 (5.6) | 4 (16.7) | 20 (83.3) | 0.411 |
| | | No | 403 (94.4) | 45 (11.2) | 358 (88.8) | |
| **Preeclampsia Complication** | 459 | Yes | 54 (11.8) | 7 (13.0) | 47 (87.0) | 0.855 |
| | | No | 405 (88.2) | 49 (12.1) | 356 (87.9) | |
| **Anaemia** | 413 | Yes | 139 (33.7) | 18 (12.9) | 121 (87.1) | 0.023 |
| | | No | 274 (66.3) | 61 (22.3) | 213 (77.7) | |
| **Route of Delivery** | 966 | NVD | 742 (76.8) | 74 (10.0) | 668 (90.0) | 0.012 |
| | | C/S | 224 (23.2) | 36 (16.1) | 188 (83.9) | |
| **Family History of Preeclampsia** | 1018 | Yes | 26 (2.6) | 21 (80.8) | 5 (19.2) | <0.001 |
| | | No | 992 (97.4) | 103 (10.4) | 889 (89.6) | |
| **Family History of Eclampsia** | 555 | Yes | 24 (4.3) | 18 (75.0) | 6 (25.0) | <0.001 |
| | | No | 531 (95.7) | 54 (10.2) | 477 (89.8) | |
| **Eclampsia** | 563 | Yes | 31 (5.5) | 16 (51.6) | 15 (48.4) | <0.001 |
| | | No | 532 (94.5) | 58 (10.9) | 474 (89.1) | |
| **Fam. History of Chronic HTN** | 602 | Yes | 75 (12.5) | 26 (34.7) | 49 (65.3) | <0.001 |
| | | No | 527 (87.5) | 50 (9.5) | 477 (90.5) | |
| **Diabetes Mellitus** | 700 | Yes | 19 (2.7) | 5 (26.3) | 14 (73.7) | 0.067 |
| | | No | 681 (97.3) | 83 (12.2) | 598 (87.8) | |
| **History of Smoking** | 729 | Yes | 18 (2.5) | 2 (11.1) | 16 (88.9) | 0.845 |
| | | No | 711 (97.5) | 90 (12.7) | 621 (87.3) | |
| **Current Smoking** | 789 | Yes | 7 (0.9) | 0 (0.0) | 7 (100.0) | 0.306 |
| | | No | 782 (99.1) | 95 (12.1) | 687 (87.9) | |
| **History of Alcohol** | 736 | Yes | 98 (13.3) | 21 (21.4) | 77 (78.6) | 0.005 |
| | | No | 638 (86.7) | 72 (11.3) | 566 (88.7) | |

**Abbreviation:** NVD; Normal vaginal delivery, C/S; Caesarean section, HELLP; Haemolysis Elevated Liver enzymes and Low Platelets, HTN; Hypertension.

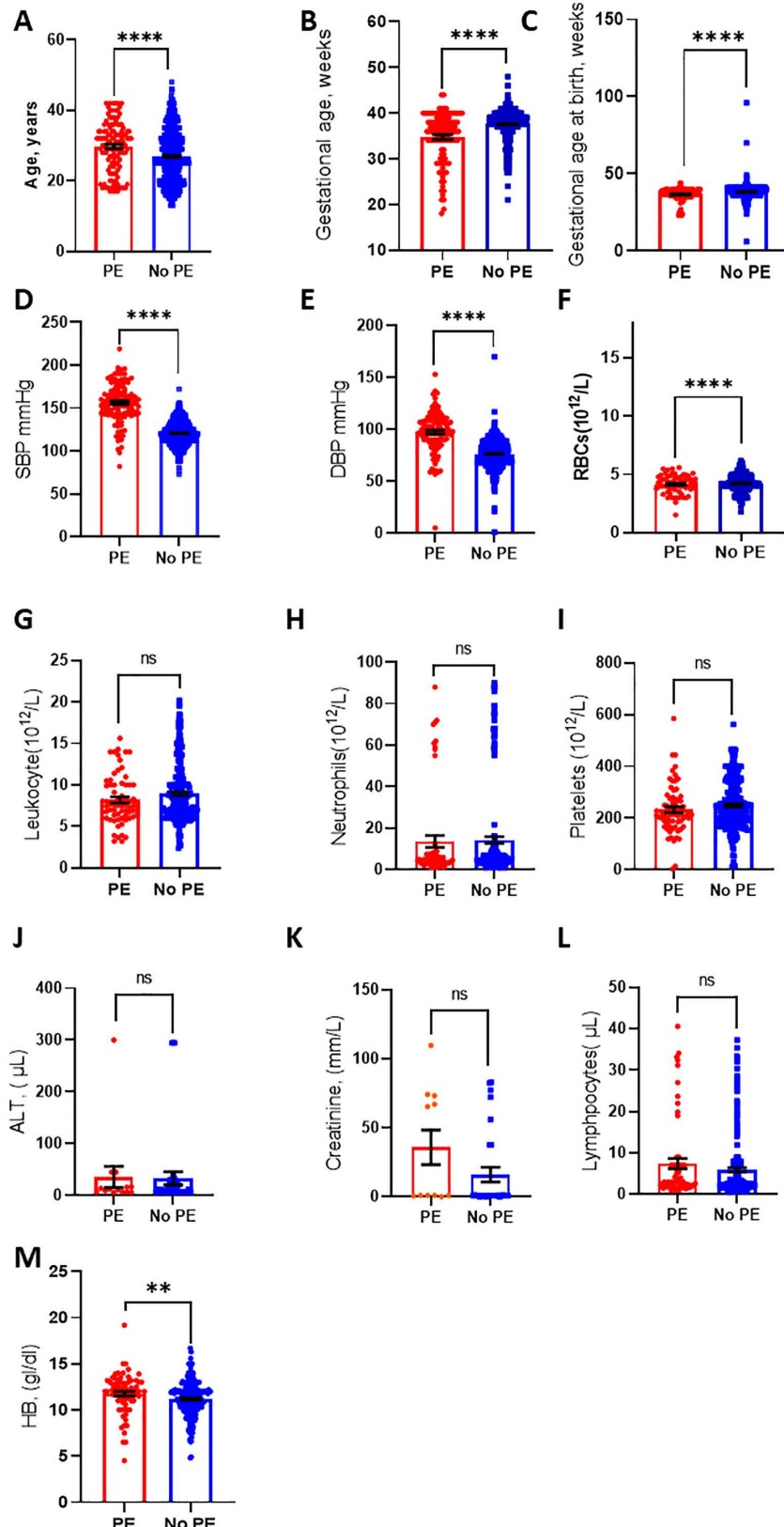

**Fig 2. Laboratory and clinical characteristics between females with and without PE.**

**Table 2. Univariable and multivariable analysis of factors associated with Preeclampsia.**

| Variables | Univariable analysis | | Multivariable analysis | |
|---|---|---|---|---|
| | OR (95% CI) | P value | AOR (95%CI) | P value |
| Age, years | 1.05 (1.02, 1.08) | < 0.001 | 1.07 (1.02, 1,12) | **0.002** |
| Employed | 1.94 (1.25, 3.01) | 0.003 | | |
| Urban residence | 0.52(0.31, 0.86) | 0.011 | | |
| Married | 1.79 (1.15, 2.80) | 0.01 | | |
| Haemoglobin | 1.16 (1.01, 1.34) | 0.03 | | |
| Anaemia | 0.51 (0.29, 0.91) | 0.024 | | |
| Gestational Age | 0.86 (0.82, 0.89) | < 0.001 | 0.89 (0.83, 0.97) | **0.007** |
| Fetal adverse outcomes | 1.89 (1.07, 3.35) | 0.026 | | |
| Gravidity | 1.18 (1.08, 1.30) | 0.0002 | | |
| Parity | 1.15 (1.05, 1.27) | 0.002 | | |
| Family History of preeclampsia | 36.2 (13.38, 98.18) | < 0.001 | 9.97 (2.53, 39.27) | **< 0.001** |
| Eclampsia | 8.71 (4.09, 18.55) | < 0.001 | | |
| Family History of Chronic hypertension | 5.06 (289, 88.4) | < 0.001 | | |
| History of alcohol intake | 2.14 (1,24, 3.86) | 0.005 | 2.49 (0.99, 6.27) | 0.051 |
| History of preeclampsia | 28.2 (12.6, 63.0) | < 0.001 | 30.8 (8.7, 108.6) | **< 0.001** |
| HELLP syndrome | 14. 21(1.26, 159.33) | 0.031 | | |
| Delivery by C/S | 1.72 (1.12, 2.65) | 0.012 | | |

*Abbreviations: OR; Odds ratio, AOR; Adjusted odds ratio, C/S Caesarean section, HELLP; Hemolysis Elevated liver enzyme low platelets*

In multivariable analysis, as age increases the odds of having PE increased by 1.07 times (AOR: 1.07: 95%CI: 1.02, 1.12 p=0.002). An increase in gestational age reduced the odds of having PE by 11% (AOR: 0.89: 95%CI: 0.83, 0.97 p=0.007). Additionally, those with a family history of PE had 9.97 times the odds of having PE (AOR: 9.97: 95%CI: 2.53, 39.27 p=0.001). Furthermore, those with a history of PE had 30.8 times the odds of developing PE (AOR: 30.8: 95%CI: 8.7, 108.6: p<0.001).

## Discussion

This study aimed to identify the correlates of pre-eclampsia (PE) among pregnant women at Livingstone University Teaching Hospital. The prevalence of PE in our cohort was 12.2%. This rate is substantially higher than the global average of 2–8% [14] and the rates reported in Congo, Tanzania, and Nigeria (1.4–4.8%) [15,16]. However, it aligns closely with rates found in other sub-Saharan African settings, such as Ethiopia (8.4–12.4%) and South Africa (12.5%) [6,14], and is slightly higher than the 12% reported by Mukosha et al. in Zambia [6]. The significant heterogeneity in PE prevalence across studies can be attributed to regional disparities in risk factor profiles, socioeconomic determinants of health, and methodological variations, including differences in diagnostic criteria and study design [17,18]. Our study's relatively large sample size (N=1,018) enhances the precision of our estimate and may partially explain the higher prevalence compared to studies with smaller cohorts. Furthermore, the inherently idiopathic and multifactorial nature of PE's etiology, involving complex gene-environment interactions that are not yet fully elucidated, contributes to the wide variability observed across different populations.

Gestational age emerged as a significant correlate, with earlier gestation being strongly associated with PE development. This finding is consistent with a robust body of literature [17,19] and is underpinned by distinct pathophysiological pathways. Early-onset PE (diagnosed before 34 weeks) is predominantly characterized by severe placental dysfunction. The primary insult is a failure of trophoblast invasion during the first and early second trimester, leading to inadequate spiral artery remodelling. This results in persistently narrow, high-resistance vessels, causing placental hypoperfusion, oxidative stress, and hypoxia [20]. The ischemic placenta responds by releasing a surge of anti-angiogenic factors, primarily

soluble Fms-like tyrosine kinase-1 (sFlt-1), and inflammatory cytokines into the maternal circulation [21]. sFlt-1 binds to and depletes circulating placental growth factor (PlGF) and vascular endothelial growth factor (VEGF), culminating in widespread maternal endothelial dysfunction, systemic vasoconstriction, and the clinical syndrome of PE [21,22]. This mechanism explains the strong association with preterm delivery and underscores the critical need for vigilant monitoring and PlGF-based biomarker screening in early pregnancy to identify at-risk women.

A prior history of PE was the strongest predictor of the condition in our study, highlighting a profound recurrence risk. This risk may be explained by shared genetic and epigenetic factors, such as abnormal DNA methylation that alters gene expression [23,24]. Persistent vascular damage and endothelial dysfunction may also contribute [25,26], while coexisting chronic conditions like hypertension, diabetes, and obesity further elevate risk [27,28]. In addition, immune maladaptation has been implicated, as women with prior PE often exhibit heightened inflammatory responses or reduced tolerance to novel paternal antigens in subsequent pregnancies [29,30].

Similarly, a family history of PE was a powerful predictor (AOR: 9.97), underscoring the role of genetic predisposition [31,32]. Evidence from the *American Journal of Epidemiology* indicated that a family history, particularly among female relatives, markedly increases the likelihood of early-onset PE [33,34]. Similarly, Wu et al. demonstrated that maternal and sibling histories of PE or hypertension are linked to severe PE [35], and Meazaw et al.'s systematic review confirmed family history as a major predictor [11]. The strong familial aggregation, particularly through maternal lineages [35,36], points to a heritable component involving polymorphisms in genes regulating blood pressure, angiogenesis, and placental development [10,37]. Beyond genetics, epigenetic modifications provide a key mechanistic link for recurrence risk. Aberrant DNA methylation patterns in genes critical for vascular and placental function can be acquired from a prior affected pregnancy and persist, predisposing to endothelial dysfunction in subsequent pregnancies [38,39]. This "vascular priming" effect means that pre-existing subclinical vascular damage from a previous PE episode can be exacerbated by the physiological stresses of a new pregnancy.

Given the strength of these associations, our findings have direct clinical implications. Incorporating a thorough assessment of obstetric history such as previous preeclampsia and fetal growth restriction, together with family history in first-degree relatives, into antenatal risk stratification is both prudent and essential. This cost-effective approach allows early identification of high-risk women during the first trimester, enabling timely initiation of prophylactic measures such as low-dose aspirin and facilitating closer monitoring to improve outcomes.

**Study limitations and strengths.** The major strengths of this study include its robust sample size of 1,018 participants and standardized dual-verified data collection procedures, which enhance the methodological rigor and reliability of the findings. This study offers important contributions to understanding preeclampsia (PE) correlates while acknowledging certain methodological limitations. The analysis examined an extensive range of variables spanning demographic characteristics, clinical factors, and laboratory measures, enabling thorough adjustment for potential confounders in multivariable models. Importantly, this work provides valuable data on PE risk factors in Zambia, where such research remains limited despite the condition's substantial burden on maternal health outcomes, helping to address critical gaps in our understanding of PE epidemiology in sub-Saharan African populations.

Nonetheless, certain limitations should be considered. The retrospective design means findings depend on the completeness and accuracy of existing medical records, which could introduce some degree of information bias. Conducted at a single tertiary care facility, the results may not fully extend to other healthcare settings with different patient populations or resource levels. The retrospective data collection relied on existing records, which may not account for potential seasonal variations in PE cases. The use of dipstick proteinuria testing, though clinically convenient, may be less precise than laboratory-based quantitative methods. As an observational study, it cannot establish causal relationships, and the lack of information on dietary factors, physical activity, and environmental influences leaves some questions unanswered about potential modifying effects.

## Conclusion

Age, gestational age, history of PE and a family history of PE were strongly associated with PE. These findings support the integration of targeted screening tools into maternal health programs and underscore the need for more prospective cohort studies to explore causality and intervention strategies.

## Supporting information

**S1 File. Strobe checklist.**
(DOCX)

**S2 File. Data Set.**
(CSV)

## Author contributions

**Conceptualization:** Martin Chakulya, Annet Kirabo, Sepiso K. Masenga.

**Data curation:** Martin Chakulya, Gift C. Chama, Bislom C. Mweene, Annet Kirabo, Sepiso K. Masenga.

**Formal analysis:** Martin Chakulya, Gift C. Chama, Lukundo Siame, Sepiso K. Masenga.

**Investigation:** Martin Chakulya, Sepiso K. Masenga.

**Methodology:** Martin Chakulya, Annet Kirabo, Sepiso K. Masenga.

**Software:** Isaac Moonga Pulukuta.

**Supervision:** Martin Chakulya, Annet Kirabo, Sepiso K. Masenga.

**Validation:** Martin Chakulya, Prince Mulambo, Lillian nalavwe, Isaac Moonga Pulukuta, Portipher Silwamba, Wana Nyichiwu, Alice Mwape, Sydney Mulamfu, Katongo Hope Mutengo, Lukundo Siame, Joreen P. Povia, Bridget Namusiku, Bislom C. Mweene, Annet Kirabo, Sepiso K. Masenga.

**Visualization:** Martin Chakulya, Prince Mulambo, Gift C. Chama, Lillian nalavwe, Isaac Moonga Pulukuta, Portipher Silwamba, Enniless Chilobe, Wana Nyichiwu, Alice Mwape, Emmanuel Luwaya, Sydney Mulamfu, Chileleko Siakabanze, Katongo Hope Mutengo, Lukundo Siame, Joreen P. Povia, Bridget Namusiku, Bislom C. Mweene, Annet Kirabo, Sepiso K. Masenga.

**Writing – original draft:** Martin Chakulya, Prince Mulambo, Gift C. Chama, Lillian nalavwe, Portipher Silwamba, Enniless Chilobe, Wana Nyichiwu, Alice Mwape, Emmanuel Luwaya, Sydney Mulamfu, Chileleko Siakabanze, Katongo Hope Mutengo, Lukundo Siame, Joreen P. Povia, Bridget Namusiku, Bislom C. Mweene, Annet Kirabo, Sepiso K. Masenga.

**Writing – review & editing:** Martin Chakulya, Prince Mulambo, Gift C. Chama, Lillian nalavwe, Portipher Silwamba, Enniless Chilobe, Wana Nyichiwu, Alice Mwape, Emmanuel Luwaya, Sydney Mulamfu, Chileleko Siakabanze, Katongo Hope Mutengo, Lukundo Siame, Joreen P. Povia, Bridget Namusiku, Bislom C. Mweene, Annet Kirabo, Sepiso K. Masenga.

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
