## [Decision Letter · Decision Letter 0]

16 Aug 2025

Dear Dr. CHAKULYA,

**This study effectively identifies factors associated with preeclampsia in Zambia, employing satisfactory methodology and analysis. This study provides valuable indigenous data on preeclampsia factors in Zambia. **

**Background Queries**

• Include a sentence that emphasises the research gap that relates with the regional context (e.g., Sub-Saharan Africa or your study’s setting), to justify the importance of the study.

• Consider the inclusion of recent citations (preferably within the last 5 years), to emphasise significance (e.g., [1], [2])

• There is insufficient reference to empirical population data, particularly for low- and middle-income countries (LMICs), despite acknowledging their burden: Revise

**Methods Queries**

• Several grammatical errors are evident, e.g., "form" instead of "from": Revise accordingly throughout

• Clarify the systematic sampling method

• Include the sample size calculation formula

**Results Queries**

• Whilst the data is adequately presented, a narrative synthesis is required to validate the significance of the key findings

• Some statements ambiguous, for example, “An increase in gestational age increased the odds of having PE by 11% (OR: 0.86...)” - This is contradictory, OR <1 indicates decreased odds, not increased.

• Explain the observations noted in Figure 2. For example: "Women with PE had significantly lower gestational age and higher blood pressure, while hematological parameters did not significantly differ.": Why?

**Discussion and Conclusion Queries**

• Provide additional information regarding the strength of associations (e.g., AORs of 9.97 and 30.8); the clinical relevance of variables (e.g., why should family history be included in all antenatal intake forms?).

• Explain how the data can inform antenatal screening or referrals

Strengthen the conclusion by including the clinical relevance of the study findings.

**References**

Please refer to the PLOS ONE Referencing guidelines and revise accordingly.

We look forward to receiving your revised manuscript.

Kind regards,

Nalini Govender, Ph.D

Academic Editor

PLOS ONE

**Journal Requirements:**

1. When submitting your revision, we need you to address these additional requirements. Please ensure that your manuscript meets PLOS ONE's style requirements, including those for file naming. The PLOS ONE style templates can be found at https://journals.plos.org/plosone/s/file?id=wjVg/PLOSOne_formatting_sample_main_body.pdf and https://journals.plos.org/plosone/s/file?id=ba62/PLOSOne_formatting_sample_title_authors_affiliations.pdf 2. Please amend either the abstract on the online submission form (via Edit Submission) or the abstract in the manuscript so that they are identical. 3. We note that there is identifying data in the Supporting Information file. Due to the inclusion of these potentially identifying data, we have removed this file from your file inventory. Prior to sharing human research participant data, authors should consult with an ethics committee to ensure data are shared in accordance with participant consent and all applicable local laws. Data sharing should never compromise participant privacy. It is therefore not appropriate to publicly share personally identifiable data on human research participants. The following are examples of data that should not be shared: -Name, initials, physical address-Ages more specific than whole numbers-Internet protocol (IP) address-Specific dates (birth dates, death dates, examination dates, etc.)-Contact information such as phone number or email address-Location data-ID numbers that seem specific (long numbers, include initials, titled “Hospital ID”) rather than random (small numbers in numerical order) Data that are not directly identifying may also be inappropriate to share, as in combination they can become identifying. For example, data collected from a small group of participants, vulnerable populations, or private groups should not be shared if they involve indirect identifiers (such as sex, ethnicity, location, etc.) that may risk the identification of study participants. Additional guidance on preparing raw data for publication can be found in our Data Policy (https://journals.plos.org/plosone/s/data-availability#loc-human-research-participant-data-and-other-sensitive-data) and in the following article: http://www.bmj.com/content/340/bmj.c181.long. Please remove or anonymize all personal information (<specific identifying information in file to be removed>), ensure that the data shared are in accordance with participant consent, and re-upload a fully anonymized data set. Please note that spreadsheet columns with personal information must be removed and not hidden as all hidden columns will appear in the published file. 4. Please include captions for your Supporting Information files at the end of your manuscript, and update any in-text citations to match accordingly. Please see our Supporting Information guidelines for more information: http://journals.plos.org/plosone/s/supporting-information. 5. If the reviewer comments include a recommendation to cite specific previously published works, please review and evaluate these publications to determine whether they are relevant and should be cited. There is no requirement to cite these works unless the editor has indicated otherwise. 

**Additional Editor Comments:**

Thank you for your submission. This study effectively identifies factors associated with preeclampsia in Zambia, employing satisfactory methodology and analysis. Future research exploring causal relationships could enhance preventive strategies, benefiting Zambia, and Sub-Saharan Africa, where preeclampsia's disease burden is substantial. This study provides valuable indigenous data on preeclampsia factors in Zambia. Future collaborative research in Sub-Saharan Africa, focusing on causal associations, could inform prevention strategies and help mitigate maternal and perinatal morbidity and mortality in the region.

However, the following queries need to be addressed before it can be considered for publication. Please carefully revise the manuscript by responding to all the reviewers' comments, clarifying methodological issues, ensuring appropriate interpretation of the results, and improving the overall structure and grammar of the text.

**Background**

The background is well-conceived, and clearly introduces preeclampsia (PE) as a global maternal health concern and outlines its clinical definition and consequences, emphasizing the multifactorial nature of PE. The relevance of genetic, immunological, environmental, and lifestyle factors is well-articulated and aligns with current literature.

**Queries**

•     Include a sentence that emphasises the research gap that relates with the regional context (e.g., Sub-Saharan Africa or your study’s setting), to justify the importance of the study.

•     Consider the inclusion of recent citations (preferably within the last 5 years), to emphasise significance (e.g., [1], [2])

•     There is insufficient reference to empirical population data, particularly for low- and middle-income countries (LMICs), despite acknowledging their burden: Revise

**Language and Grammar**

•     Line 1: “fatal health” should be “fetal health”

•     Line 2: “may lead to rapid development of severe complications” consider “can rapidly progress to severe complications”

•     Line 4: “where access to specialized care often falls short” revise to “where access to specialized obstetric care is often limited”

•     Line 6: “external contributors like lifestyle choices” ,replace with “modifiable factors such as lifestyle behaviors”

•     Line 9: “emphasize the hereditary and epigenetic dimensions” consider “highlighting hereditary and epigenetic contributions”

•     Line 13: “dissecting these complex interactions”, could be simplified to “understanding these interactions”

**Methods**

**Revise based the PLOS ONE Guidelines**

•     Several grammatical errors are evident, e.g., "form" instead of "from", "patient’s" instead of "patients'"): Revise accordingly throughout

•     Clarify the systematic sampling method: were the files ordered or randomized?.

•     Include the sample size calculation formula, Provide more clarity on assumptions used (confidence level, power, design effect).

•     Group variables into categories (sociodemographic, clinical, laboratory, etc.) to improve readability and reduce repetition.

•     Data collection period should be checked for temporal consistency (dates should make sense in relation to the manuscript’s submission timeline).

•     Use clear definitions with numbered citations and paragraph breaks

**Results**

**Queries**

1. Grammatical, punctuation, and formatting issues throughout the document

•     For example: “form different health facilities” should be “from different health facilities”

“Unavailable and multivariable analysis” should be “Univariable and multivariable analysis”

"21.4" in “21 (21,4)” likely a formatting error; use “21 (21.4%)”

Typographic inconsistencies like “ALT; Alanine transferase” and general spacing errors (e.g., “9.97 95%Cl”) should be corrected.

2. Data Presentation

•     Whilst the data is adequately presented, a narrative synthesis is required to provide a more meaningful description. Despite the extensive statistics provided, more emphasis is needed to validate the significance of the key findings.

•     There are inconsistencies in the Figures: Eg.  For "Haemoglobin, gram/dl", it is shown as “31 (25, 34)”: Check throughout and revise where necessary .

•     Consider renaming the variable "Gestational_age_weeks_2" to increase the clarity in the manuscript.

•     Multivariable Model Reporting: Provide a brief explanation regarding the selection of variables from the univariable analysis for inclusion in the adjusted multivariable model (e.g., p-value < 0.2 threshold or stepwise regression?).

•     Ensure the formatting is consistent throughout the document:

o     p-values (e.g., "p < 0.001", not "P=0.0001" or “p<0.0.1”)

o     Units (e.g., "g/dL", "µmol/L", "x10³/µL")

**Statistical Accuracy and Interpretation**

•     Some statements ambiguous, for example,  “An increase in gestational age increased the odds of having PE by 11% (OR: 0.86...)” - This is contradictory,  OR <1 indicates decreased odds, not increased.

•     Check the use of terms "increase" vs. "decrease" when referring to the direction of ORs.

•     Explain in detail the observation noted in Figure 2.  For example: "Women with PE had significantly lower gestational age and higher blood pressure, while hematological parameters did not significantly differ.": Why?

**Discussion**

Despite the well formulated discussion, the following queries must be addressed:

•     Evaluate the strength of associations (e.g., AORs of 9.97 and 30.8); the clinical relevance of variables (e.g., why should family history be included in all antenatal intake forms?).

•     Minor grammatical issues were observed, e.g., “unexplained and idiopathic explaining why”;   repetitive phrasing  of “another variable that emerged…”.

•     Rephrase sentences to demonstrate more clarity: For example Revise “Our study revealed a slightly higher prevalence (12.7%) than Mukosha et al. (12%), although both findings are largely consistent.”

•     Avoid redundancy: “history of PE was significantly associated with recurrence” and “prior PE are at increased risk” can be merged.

•     Break long paragraphs into smaller ones to improve readability, especially the third paragraph.

•     Provide more evidence on the public health and/or clinical implications.

o     For example: Explain how the data can inform antenatal screening or referrals

•     Move the sentence on sample size and data verification closer to the beginning of the paragraph to maintain balance.

**Conclusion**

Consider revising by strengthening the  clinical relevance of the study.

**References**

Please refer to the PLOS ONE Referencing guidelines and revise accordingly. For example, no italics or bold, no quotation marks, and no brackets around volume or page numbers.

**Queries**

•     Remove Duplicate References

•     Use full journal titles in title case, not abbreviations or partial capitalizations.

•     Ensure Consistent Author Formatting; List up to 6 authors, then use et al. after the 6th.

Reviewers' comments:

Reviewer's Responses to Questions

**Comments to the Author**

1. Is the manuscript technically sound, and do the data support the conclusions?

Reviewer #1: Yes

Reviewer #2: Yes

2. Has the statistical analysis been performed appropriately and rigorously?

Reviewer #1: Yes

Reviewer #2: Yes

3. Have the authors made all data underlying the findings in their manuscript fully available?

Reviewer #1: Yes

Reviewer #2: Yes

4. Is the manuscript presented in an intelligible fashion and written in standard English?

Reviewer #1: Yes

Reviewer #2: Yes

**Reviewer #1: ** This study effectively identifies factors associated with preeclampsia in Zambia, employing satisfactory methodology and analysis. Future research exploring causal relationships could enhance preventive strategies, benefiting Zambia and Sub-Saharan Africa, where preeclampsia's disease burden is substantial.

**Reviewer #2: ** Thank you for your submission. While your manuscript presents important findings, several concerns and queries need to be addressed before it can be considered for publication. Please carefully revise the manuscript by responding to all the reviewers' comments, clarifying methodological issues, ensuring appropriate interpretation of the results, and improving the overall structure and grammar of the text.

**Do you want your identity to be public for this peer review?** For information about this choice, including consent withdrawal, please see our Privacy Policy

Reviewer #1: **Yes: ** Dr Revathi Soundararajan

Reviewer #2: No

---

## [Author Response · Author response to Decision Letter 1]

7 Oct 2025

29/09/2025

PLOS ONE Journal

Dear Editor,

Ref: Submission of a revised research article for peer review and publication consideration

Reference to the above-mentioned subject. I am writing to submit a revised original research article titled "Preeclampsia, prevalence and associated factors,". We would like to thank the reviewers for taking the time to make suggestions that have improved our manuscript. We have revised the manuscript and addressed all concerns and suggestions. We now hope the current manuscript is acceptable for publication. Below are the point-by-point responses to all comments and suggestions

Response to the reviewers

Background Review - Scientific Rigor and Relevance

Reviewer: Specificity of sources: While references are cited, clarity in source attribution is important. More precise statements referencing current and key studies would enhance credibility.

Response: We thank the reviewer for this critical observation. We have thoroughly revised the background section to incorporate more specific and recent references. Key statements on the global and regional burden of PE have been updated with precise citations from recent studies (within the last 5 years where available) to strengthen the credibility and context of our work [Updated Refs 2-6]. We have also ensured that citations for the clinical definition [Ref 1] and pathophysiological mechanisms [Refs 7-9] are clear and authoritative.

Reviewer: Balance: The section leans heavily on theoretical associations but lacks sufficient grounding in empirical population data for LMICs.

Response: We agree. We have revised the background to better contextualize known risk factors within the LMIC and Zambian setting. We have added a sentence to explicitly state the scarcity of comprehensive data from regions like Southern Zambia and have strengthened the rationale for our study by highlighting that it aims to provide this missing empirical, population-based evidence from a high-burden setting to inform local clinical practice and policy.

Reviewer: Language and Grammar corrections (multiple specific examples provided).

Response: We have incorporated all these language and grammar corrections into the revised manuscript to improve clarity, precision, and academic tone.

Methods Section

Reviewer: Grammar and syntax corrections are needed.

Response: Thank you. We have revised the grammar and syntax throughout the methods section.

Reviewer: Sample size calculation requires more clarity on assumptions.

Response: Thank you. We have clarified the sample size calculation by explicitly stating the parameters used: an expected frequency of 7.7%, a 95% confidence level (Z=1.96), a margin of error of 5%, and a design effect of 1. The formula has been included in the text for transparency.

Reviewer: Systematic sampling method needs clarification.

Response: Thank you. We have clarified the sampling strategy. The list of patient records was ordered by date of visit. The starting point was then randomly selected before selecting every third file, which is a standard systematic sampling approach.

Reviewer: Grouping of variables will improve readability.

Response: Thank you. We have now grouped the independent variables into clear categories: Sociodemographic, Obstetric and reproductive history, Clinical, and Laboratory haematological variables.

Reviewer: Data collection period should be checked for temporal consistency.

Response: We sincerely apologize for this critical error. The correct data collection period was 1st December 2023 to 30th March 2024. This has been corrected in the manuscript.

Results Section

Reviewer: Language and Grammar corrections (multiple specific examples provided).

Response: Thank you. We have revised all the indicated grammar, punctuation, and formatting issues for clarity and professionalism (e.g., "from different health facilities," "Univariable," "21.4%," consistent spacing and units).

Reviewer: The results would benefit from more narrative synthesis.

Response: We have added a stronger narrative summary at the beginning of the results section to better highlight and interpret the key findings in context.

Reviewer: Multivariable Model Reporting: Please state how variables were selected for inclusion.

Response: We have added the following explanation to the methods section: "Variables with a p-value < 0.01 in the univariable analysis, along with those of known clinical importance, were considered for inclusion in the multivariable logistic regression model. We also excluded variables with significant collinearity. A backward stepwise selection process was used to arrive at the final adjusted model."

Reviewer: Statistical Accuracy: Contradictory statement regarding gestational age odds ratio.

Response: Thank you for identifying this error. We have corrected the narrative to accurately reflect that an increase in gestational age is associated with decreased odds of PE (OR < 1).

Discussion and Conclusion

Reviewer: While findings are well-discussed, it could benefit from a more analytical tone evaluating the strength of associations and clinical relevance.

Response: We have revised the discussion to more critically analyze the strength of our high odds ratios (e.g., AOR of 30.8 for prior history) and their profound clinical relevance. We have added a specific recommendation on integrating family and obstetric history into antenatal risk stratification.

Reviewer: Language and Writing Quality: Minor grammatical errors and repetitive phrasing.

Response: Thank you. We have corrected the grammatical errors (e.g., "unexplained and idiopathic"), eliminated repetitive phrasing, broken up long paragraphs, and improved overall clarity.

Reviewer: Limited commentary on public health or clinical implications.

Response: We have added a sentence in the discussion to address this: "Incorporating routine assessment of prior and family history of PE into antenatal risk stratification may facilitate early detection and reduce adverse maternal outcomes."

Reviewer: Conclusion: Make it slightly more actionable.

Response: We have revised the conclusion as suggested: "These findings support the integration of targeted screening tools into maternal health programs and underscore the need for more prospective cohort studies to explore causality and intervention strategies."

Reference Section

Reviewer: Remove duplicate references and correct journal name formatting.

Response: Thank you. We have removed the duplicate entries for Karrar et al. and Mukosha et al. We have also reformatted all journal titles to their full names (not abbreviated) and ensured consistency with the Vancouver/PLOS ONE style throughout the reference list.

---

## [Editor Report · Decision Letter 1]

6 Nov 2025

Preeclampsia, prevalence and associated factors

PONE-D-25-21925R1

Dear Dr. Chakulya,

We’re pleased to inform you that your manuscript has been judged scientifically suitable for publication and will be formally accepted for publication once it meets all outstanding technical requirements.

Kind regards,

Nalini Govender, Ph.D

Academic Editor

PLOS ONE
---

## [Editor Report · Acceptance letter]

PONE-D-25-21925R1

PLOS ONE

Dear Dr. Chakulya,

I'm pleased to inform you that your manuscript has been deemed suitable for publication in PLOS ONE. Congratulations! Your manuscript is now being handed over to our production team.

Kind regards,

on behalf of

Prof Nalini Govender

Academic Editor

PLOS ONE